# Let’s Get Physical: Bacterial-Fungal Interactions and Their Consequences in Agriculture and Health

**DOI:** 10.3390/jof6040243

**Published:** 2020-10-23

**Authors:** Breanne N. Steffan, Nandhitha Venkatesh, Nancy P. Keller

**Affiliations:** 1Department of Medical Microbiology and Immunology, University of Wisconsin-Madison, Madison, WI 53706, USA; bsteffan@wisc.edu (B.N.S.); thiruvannama@wisc.edu (N.V.); 2Department of Plant Pathology, University of Wisconsin-Madison, Madison, WI 53706, USA; 3Department of Bacteriology, University of Wisconsin-Madison, Madison, WI 53706, USA

**Keywords:** bacterial–fungal interactions, adhesion, attachment, endofungal

## Abstract

Fungi serve as a biological scaffold for bacterial attachment. In some specialized interactions, the bacteria will invade the fungal host, which in turn provides protection and nutrients for the bacteria. Mechanisms of the physical interactions between fungi and bacteria have been studied in both clinical and agricultural settings, as discussed in this review. Fungi and bacteria that are a part of these dynamic interactions can have altered growth and development as well as changes in microbial fitness as it pertains to antibiotic resistance, nutrient acquisition, and microbial dispersal. Consequences of these interactions are not just limited to the respective microorganisms, but also have major impacts in the health of humans and plants alike. Examining the mechanisms behind the physical interactions of fungi and bacteria will provide us with an understanding of multi-kingdom community processes and allow for the development of therapeutic approaches for disease in both ecological settings.

## 1. Introduction

Fungi and bacteria coexist creating complex communities that are important in agriculture and human health. Although originally mostly studied in agricultural environments (e.g., arbuscular mycorrhizal-bacterial [1] and *Rhizopus-Mycetohabitans* (formerly *Burkholderia* [2]) symbioses [3]), bacterial–fungal interactions (BFI) are now recognized as the norm in most ecological settings. In fact, identifying microorganisms that make up these communities has been the focus of numerous studies such as the Human Microbiome Project started by the National Institutes of Health (NIH) in 2007 [4].

BFIs are commonly studied in biofilms, where both microbes physically interact, often exchanging or responding to each other’s metabolites. Studying their direct physical interactions informs researchers of their collective behavior on survival, production of secondary metabolites (that find use as clinical drugs, fungicides, prebiotics etc.), dissemination of the organisms to different environments, and identification of mechanisms that facilitate these types of interactions. As we will show in this review, direct physical interactions of bacteria and fungi typically present with fungi as scaffolds where bacteria can be external or internal to the fungus. These physical interactions can be dependent on the microbial partner, the host (if any) of the BFI, and even the substrate for growth. We will address the findings across practices—in agriculture and clinical settings—to call attention to a unified outlook on the ecology of how microbes interact. We propose that the combined insight with respect to microbial ecology can aid in obtaining a better understanding of clinical and agricultural outcomes and help to identify novel methods for disease control and the promotion of good health of plants as well as humans.

## 2. Bacterial-Fungal Physical Interactions

Bacteria can be associated with fungi (externally and internally) and the type of interaction depends not only on the morphology of the fungal organisms but also on fungal and bacterial surface molecules and secreted factors. Most but not all fungi form extensive filaments called hyphae that are penetrative into the surrounding environment, be it the rhizosphere, organic matter, or a living host. While many BFIs are characterized by bacteria interacting with hyphal and fungal spore surfaces, there is growing evidence of bacterial invasion into fungal structures. Mechanisms of bacterial attachment to fungal surfaces and their invasion of fungal organisms are highlighted in this section (Table 1).

### 2.1. Bacterial Attachment to the External Surface of Fungi

Both clinical and agricultural settings have made great strides in addressing the mechanisms associated with bacterial attachment to fungal surfaces. Surface molecules and secreted factors drive the interactions between bacteria and fungi, though the response seems to be specific to the respective microorganisms. In this subsection, we will discuss factors contributing to the formation of BFIs across clinical and agricultural disciplines.

#### 2.1.1. Human Pathogenic Fungi and Their Bacterial Interactions

*Candida albicans* is a human commensal that can, in immunocompromised patients or in situations of microbiome dysbiosis, cause candidiasis. It is a pleomorphic fungus with yeast and hypha being its most common morphologies. BFIs are commonly formed between *Candida* spp. and several types of bacteria. *Staphylococcus*, *Streptococcus*, and *Pseudomonas* are three genera commonly associated with *C. albicans* where they adhere preferentially to the hypha and not the yeast cell [5,6,7,8,9]. Adherence of the bacteria to the hypha is dependent on fungal and bacterial surface molecules and secreted factors.

Adhesins are a class of surface molecules that facilitate bacterial attachment to the fungal hypha. Deletion of adhesin molecules are associated with reduced bacterial attachment to *C. albicans* by bacteria including *Staphylococcus epidermidis* [8], *Staphylococcus aureus* [10,11], and *Streptococcus gordonii* [9]. Beaussart et al. identified both agglutinin-like sequence (Als) proteins and *O*-mannosylations as important factors involved in *S. epidermidis* attaching to *C. albicans* [8]. The *C. albicans* adhesin agglutinin-like sequence 3 (Als3p) is implicated in the adherence of the bacteria, *S. aureus* and *S. gordonii*, to the fungus [9,10,11]. Als3p is a hyphal-specific glycoprotein that facilitates biofilm formation and binding to the host cells and extracellular matrix [9,12]. Initial work studying this adhesin in mediating BFIs found that *S. gordonii* attaches to *C. albicans* Als3p via the bacterial expression of the surface protein adhesin, SspB [9]. In similar studies, the fungal Als3p has also been found to be involved in the attachment of *S. aureus* to *C. albicans* [10,11].

Use of a *C. albicans als3* mutant demonstrated the reduced ability of *S. aureus* to bind to the fungal hypha via several tests including measuring adhesion forces, microscopy, and immunoassays [10]. As Als3p is also implicated in the attachment and invasion of *C. albicans* hyphae on mammalian tissue, mouse tongues were used ex vivo to show that, while the Δ*als3* strain hyphae were able to penetrate into the subepithelial layer, *S. aureus* was not able to adhere to the hyphae, and thus they could not be found in the subepithelium. This is in contrast with the wild-type (WT) *C. albicans*-*S. aureus* interaction in which the bacterium was found attached to invading hyphae located in the subepithelium [10]. This group later used a murine in vivo study to reiterate the importance of Als3p in facilitating the attachment of *S. aureus* to *C. albicans.* They showed that in WT *C. albicans-S. aureus* interactions, the bacteria bind to the hyphae and uses its BFI to invade and disseminate throughout host tissue resulting in greater morbidity and mortality of the mice. In contrast, the deletion of Als3p in *C. albicans* reduces *S. aureus* binding and dissemination in the host thus resulting in better survival outcomes for the animals [11]. Like the *C. albicans*-*S. gordonii* BFI, *S. aureus* attachment to *C. albicans* is also mediated by bacterial adhesins [9,11]. There were several staphylococcal mutants that showed a reduction in attachment to *C. albicans* including those lacking the fibronectin binding protein B, the surface protein SasF, and the bifunctional autolysin Atl. In this study, the adhesion of *S. aureus* to *C. albicans* was not completely abrogated, which suggests that there may be other factors that contribute to the attachment of the bacteria to hyphae in this BFI [11].

*C. albicans* has also been found to interact with the Gram-negative bacterium, *P. aeruginosa* in clinical disorders like cystic fibrosis. There are a few different mechanisms that have been found to contribute to *P. aeruginosa* binding to *C. albicans.* One such factor is in the hydrophobicity of the hyphae. As *P. aeruginosa* preferentially binds to the hyphae, one research group questioned the impact of acid-base attractive forces. They found that by reducing the hydrophobicity of the hyphae via a pronase treatment, there is a reduction in the binding of *P. aeruginosa* to *C. albicans*. This could be due in part to the loss of the mannoprotein layer [6]. The chitin-binding protein, CbpD found expressed by *P. aeruginosa* also impacts the *C. albicans–P. aeruginosa* BFI as Cbpd *P. aeruginosa* mutants show reduced adherence to the hyphae of *C. albicans.* This attachment can be restored by complementing with CbpD [5].

*P. aeruginosa* also binds to the filamentous pathogen *Aspergillus fumigatus.* Together, these organisms have been found to cause poor health outcomes in cystic fibrosis patients [13]. In a study by Briard et al. *P. aeruginosa* secreted dirhamnolipids that induced the production of an *A. fumigatus* extracellular matrix (ECM) enriched in galactosaminogalactan (GAG), 1,8-dihydroxynaphthalene-melanin, and pyo-melanin [14]. This ECM surrounds the hyphae which facilitates *P. aeruginosa* binding to the site of greatest metabolic activity on the hyphae. Biochemical investigations indicated that bacteria only bound to GAG via ionic interactions. *P. aeruginosa* also impacts the cell wall structure of *A. fumigatus* as the fungus has a thicker cell wall in its presence. Specifically, the dirhamnolipid, diRha-C_10_-C_10_, produced by *P. aeruginosa* is responsible for the increased thickness of the cell wall and is correlated to increased GAG [14]. Inducing the production of ECM material can also occur in the interactions of *C. albicans* and *S. aureus*. The matrix produced allows the bacterium to be tightly associated with the hyphae and survive in unfavorable environments like that of serum or exposure to vancomycin [15].

#### 2.1.2. Hyphal Attachment in Environmental BFIs

As with clinical BFIs, microbial partners that are in environmental BFIs have been studied to identify factors that enable their attachment, this includes fungal viability, secreted components, and surface markers. Hyphal viability influences bacterial attachment though bacteria differ in their need for live hyphae. When considering the BFI between *Glomus* sp. and bacteria, *Bacillus cereus* and *Paenibacillus peoriae* attach best to nonviable hypha, *Pseudomonas fluorescens* can bind to both living and non-living, and *Paenibacillus brasilensis* requires live hypha for attachment [16]. As we will later discuss in Section 3.2, some bacteria will bind to the hypha to exchange nutrients with fungi in otherwise adverse environments. Therefore, we can infer that bacteria, like *P. brasilensis*, interact with only live hypha to gain some competitive advantage from this BFI.

Additional studies have further characterized the movement of bacteria along the surface of hypha (termed the fungal highway—discussed in Section 3.3). In this context, dead fungal hypha may be advantageous to bacterial attachment and migration as otherwise living fungi may produce toxins that inhibit the attachment, a phenomenon that has been studied with BFIs of *Serratia marcescens* and *A. fumigatus* or *Rhizopus oryzae.* A mediator of these BFIs is the type I fimbriae which are involved in the attachment of *S. marcescens* to either *A. fumigatus* or *R. oryzae.* If *S. marcescens* has deficient fimbrial expression, it more effectively migrates across the fungal mycelia as there is a looser attachment formed between bacterium and fungus [17].

Bacterial secreted products are also important components in forming physical interactions with fungal partners. The soil dwelling bacterium, *Bacillus subtilis*, has been found to establish biofilms on the surfaces of *Aspergillus niger* and *Agaricus bisporus* mycelia. Formation of the biofilm depends on the production of bacterial secreted matrix components (exopolysaccharide (EPS) and the major biofilm matrix component (TasA) amyloid fiber) which are globally regulated by the stage 0 sporulation protein A (Spo0A). If any of these components are deleted from the genome of *B. subtilis*, the bacterium is no longer able to attach and form biofilms on the fungal surfaces. However, supplementing the matrix components back in facilitates the biofilm formation [18].

Surface attachment of bacteria to fungi is only one aspect of BFIs and it is likely important in establishing more complex interactions, including bacterial invasion into fungi, which will be discussed in Section 2.2.

### 2.2. Bacterial Invasion into Fungi

The very first observation of the endo-fungal lifestyle was reported as the presence of Bacteria-Like Organisms in *Endogone* spores [19]. It became evident from subsequent research that endo-fungal bacteria are widely present in several fungal phyla. Bacteria can be internalized into the fungal thallus-in hyphae as well as in spores. While co-evolved symbiotic bacteria and fungi have both evolved specific machinery to facilitate bacterial invasion and establishment of symbiosis, several instances of facultative endofungal associations have also been reported, whose mechanisms remain unknown. In this section, we will discuss mechanisms governing the establishment of endofungal lifestyle in both cases.

#### 2.2.1. Specialized Endofungal Symbioses

Tightly evolved endo-fungal symbioses have so far been reported between Betaproteobacteria and the fungal phyla Mucoromycota, which is suggested to include three sub-phyla: Mucoromycotina (example: *Rhizopus* spp.), Glomeromycotina (example: *Gigaspora* spp.), and Mortierellomycotina (example: *Mortierella* spp.) [20]. Commonalities and differences in genotypes and phenotypes of the subphyla and the evolution of bacterial-hosting abilities of these fungi have been discussed elsewhere [21]. Each of the examples pointed to above will be discussed in the following paragraphs in this section.

##### BFIs in Mucoromycotina

*Rhizopus microsporus*, the fungus causing rice seedling blight, serves as host to two different endobacteria, *Mycetohabitans rhizoxinica* and *M. endofungorum* [22]. A co-evolved mutualism identified between *M. rhizoxinica* and *R. microsporus* has been well studied over the years. Rhizoxin is a mycotoxin with antimitotic properties, required for the pathogenicity of the fungus on rice seedlings. In 2005, it was discovered that the toxin is produced by the endosymbiont *M. rhizoxinica* [3]. Further examination of the BFI showed that the endobacterium controls host sporulation and colonizes both fungal hyphae and conidia, thus getting transmitted vertically [23]. Compared to other members of the genus, the genome of *M. rhizoxinixa* showed a huge reduction in size, with fewer transcriptional regulators, quorum-sensing systems, and a higher number of transposons, virulence-related genes, and putative effectors. Genomic data suggested possible nutrient exchange-the bacterium consumed host metabolites while providing certain amino acids to the fungal host [24]. Since sporulation was conditional upon successful establishment of endofungal symbiosis, this bacterial-fungal pair became a handy system to identify contributing mechanisms. Holrhizin A, a linear lipopeptide has been identified to support endofungal colonization by the bacterium. Holrhizin A exhibited biosurfactant activity, altered biofilm formation and cell motility to promote colonization [25]. Work pioneered by several research groups has discovered the involvement of effectors-Type III secretion system (T3SS) [26], transcription activator-like (TAL) effector [27,28], and Type II secretion system (T2SS) [23]. Receptors that interact with these effectors and the downstream signaling pathways eliciting specific responses remain to be identified.

##### BFIs in Glomeromycotina

Arbuscular mycorrhizal (AM) fungi known to mobilize nutrients from soil, form highly branched structures called arbuscules that aid in transfer of nutrients from the fungus to the plant. These fungi can also harbor bacteria, externally or internally, in both hyphae and spores. “*Candidatus* Glomeribacter gigasporum” (CaGg) is often found in the hyphae and spores of *Gigaspora* species of AM fungi. In contrast, Mollicutes-related endobacteria (Mre) are more widely associated across AMF lineages. Both CaGg and Mre were found in the *Gigaspora margarita* spores, suggesting that more than one bacterium can co-exist endofungally [29]. The AMF-endobacteria obligate symbioses have been estimated to date back to at least when these AM fungi formed ancestral symbioses with land plants [30].

The intracellular symbiosis between *G. margarita* and the obligate endosymbiotic bacterium CaGg has been studied over the years. A remarkable number of 250,000 bacteria were estimated to be present in the same fungal cell. Electron microscopy enabled identification of the endo-bacterium in the cytoplasm [1]. The obligate nature of the AM fungus presented a challenge for in vitro studies. However, using a carrot-root system with single spore inoculations and confocal microscopy, it was identified that the endosymbiotic bacteria in *G. margarita* were vertically transmitted via asexual spores [31]. Genomic analyses of the bacterium showed a remarkably reduced genome with typical determinants of symbiosis. Transcriptional s analyses also showed that the bacterium expresses Type II, III, and IV secretion systems as well as the general secretion protein D (gspD) and the sec system at different stages of the fungal life cycle [32]. The vacB gene, typically implicated in host-cell colonization by enteroinvasive bacteria, has been implicated in the establishment of the symbiosis between *Burkholderia* sp. and *G. margarita*, suggesting conserved mechanisms of bacterial invasion across eukaryotic hosts [33].

##### BFIs in Mortierellomycotina

*Mortierella elongata*, a non-pathogenic soil fungus harbors *Mycoavidus cysteinexigens* as its obligate endo-bacterial resident [34]. Proteomics and metabolic analyses highlighted the dependence of the endo-bacterium on its fungal host for carbon and nitrogen. Under nitrogen-deplete conditions, the fungus exhibited reduced growth but had a larger endo-bacterial population [35]. Genomic analyses showed that *M. cysteinexigens* lacked the machinery typically thought to be required for host invasion by bacteria-such as chitin degradation systems and T2SS components [36].

#### 2.2.2. Facultative Endofungal Associations

In addition to obligate symbioses, BFIs with facultative endofungal behavior are also widespread. Most of such facultative associations involve Ascomycete and Basidiomycete fungi that interact with Alphaproteobacteria and Gammaproteobacteria. Like obligate symbioses, the study of facultative symbioses also began with assessment of specific interactions, but subsequent research efforts quickly showed that diverse bacteria exhibit endofungal behavior.

*Rhizobium radiobacter*, an endosymbiont of the fungus *Piriformspora indica*, contributes to the successful establishment of the symbiosis of the fungus with diverse plants [37]. From foliar endophytic fungi in Cupressaceous (Cypress family) trees, eleven endohyphal bacteria were isolated, whose genomic analyses showed absence of reduction as usually observed in obligate symbionts [38]. Although each bacterium encoded putative factors that can contribute to establishment of symbiosis, no commonalities in invasion machinery was observed. Phylogenetic analyses of diverse endohyphal bacteria from diverse foliar endophytes showed taxonomic incongruence between bacteria and their fungal hosts, suggesting a less specialized and more generalized association between the microbes [39].

In addition to endohyphal invasion, facultative associations are also observed in fungal resting spores and fruiting bodies. Ralsolamycin, a lipopeptide produced by the plant pathogen *Ralstonia solanacearum* facilitated the invasion of the producing bacterium into fungal chlamydospores [40]. The lipopeptide is hypothesized to increase the cell permeability, thus promoting invasion, much like Progidiosin, a red pigment produced by *Serratia marcescens* that increases cell membrane permeability of *Mucor irregularis*, which is proposed to facilitate invasion by the bacteria. In the latter case, the Type VI secretion system (T6SS) assembly protein TssJ and an outer membrane associated murein lipoprotein also showed significant up-regulation during the interaction process [41]. Chlamydospores of *Serendipita indica*, a biocontrol agent to fight plant disease, have traditionally harbored Rhizobia species, although recent reports have uncovered colonization by *Trinickia* spp. of bacteria [42]. Bacterial communities have been identified in ectomycorrhizal fungi and in saprotrophic mushrooms, where the community structure was dependent on the fungal phylogenetic groups. This variation has been attributed to variations in fruiting body chemistry, specifically to the C:N ratio and pH [43].

While some of these studies shed light on select mechanisms that enable specific facultative associations, the majority of the puzzle remains a mystery. Much of the molecular machinery that orchestrate endosymbiosis in specific bacterial-fungal pairs are yet to be identified. Mechanisms involved in enabling endofungal associations across diverse microbes also remain to be elucidated.

## 3. Implications of BFIs on Microbial Fitness

As can be observed from our discussion in the previous section, fungal cells can no longer be considered as a single organism, but rather as a holobiont, as has also been proposed elsewhere [44]. Fungal cells may externally and internally contain not only bacteria, but also phages and other mycoviruses which makes these eukaryotic cells a highly dynamic microcosm. In the following sections, we discuss the ecological consequences of these BFIs on the partner microbes as it pertains to microbial fitness, nutrient acquisition, and microbial dispersal along the hyphal highway (Figure 1).

### 3.1. Microbial Fitness Associated with Growth and Development of Microbial Partners

Bacterial and fungal growth and development can be fundamental to discovering new mechanisms of physical interactions as well as provide insight into the development of agonistic and antagonistic products resulting from such interactions. Some BFIs result in developmental changes that can affect the dispersal of the organisms, while others impair biofilm formation.

The clinical *A. fumigatus-P. aeruginosa* BFI demonstrates the negative impact that the BFI can have on fungal growth and biofilm development. Current studies have emphasized that *P. aeruginosa* impairs growth and biofilm formation by *A. fumigatus* [14,45,46]. Early research found that the LasIR quorum-sensing network that is involved in the production of a homoserine lactone of *P. aeruginosa* is vital to the inhibition of the fungus. *A. fumigatus* growth in the presence of WT PAO1 was greatly reduced in comparison to its growth with two PAO1 quorum sensing knockout strains (PAO1: Δ*LasR* and PAO1: Δ*LasI*). This study further found that this phenomenon also occurred with biofilm formation via indirect interactions of the bacterium and fungus [46]. The findings of this study and several others suggest that the interaction between *A. fumigatus* and *P. aeruginosa* is not just limited to the physical attachment but also in the production of compounds via indirect means that can not only affect fungal growth but also modulate the structure of the fungus [14,45,46,47]. As mentioned previously, the production of dirhamnolipids by *P. aeruginosa* promotes thicker cell walls in *A. fumigatus.* These lipids also cause stunted hyphal growth with hyperbranching and promote melanin production, which has been associated with reduced sensitivity to certain antifungals [14]. While there is an emphasis on the inhibition of *A. fumigatus* by *P. aeruginosa*, in the study by Briard et al., it was found that some bacterial volatile compounds could actually stimulate fungal growth [47]. Recently, a study reported that the secretome of *A. fumigatus* stimulates bacterial growth by altering the *P. aeruginosa* proteome [48].

The *C. albicans-P. aeruginosa* BFI is likewise impacted by products of the LasIR quorum-sensing network. It was identified that when *C. albicans* and *P. aeruginosa* are grown together, *P. aeruginosa* releases a diffusible factor, likely a homoserine lactone (3OC12HSL), that causes *C. albicans* to switch from hyphal growth to its yeast morphology. Other 12-carbon molecules are also able to inhibit *C. albicans* filamentation [49].

Establishment of successful bacterial-fungal associations, in many cases, contribute to fungal growth and/or sporulation. In the interaction between *Rhizopus microsporus* and *Mycetohabitans rhizoxinica*, fungal sporulation is contingent upon the successful establishment of symbiosis, which ensures maintenance of mutualism over generations [50]. Establishment of endofungal symbiosis between CaGg and *G. margarita* results in increased fungal sporulation [27]. In both cases, endobacteria are found in the spores, suggesting that control of host sporulation by endobacteria is sufficient to regulate their own transmission. The attachment of *Burkholderia glumae* to *F. graminearum* results in increased sporulation in vitro, which suggests that the BFI may result in enhanced dispersal of the fungus in the field [43]. Bacterial interactions with *Penicillium roqueforti* resulted in faster rate of growth and sporulation in vitro [51]. Bacteria associated with bark beetles influence growth and reproduction of their symbiotic fungi [52]. In in vitro co-cultures with the ectomycorrhizal fungus *Laccaria bicolor*, the mycorrhizal helper bacterium *Pseudomonas fluorescens* stimulates fungal radial growth, hyphal density and promotes soil colonization by the fungus [53]. Physical associations of *Methylobacterium* sp. P1-11 and *Trinickia* sp. T12-10 with *Serendipita indica* have been described in previous sections. In vitro co-cultures show increase in mycelial growth and sporulation and nutrient exchange [24]. *Rhizobium radiobacter*, the endofungal resident of *S. indica* is required for normal healthy in vitro growth and sporulation [19]. Bacterial interactions also promote growth and reproduction of numerous species of mushrooms, as reviewed elsewhere [54].

The bacterial populations that interact with fungal species can have profound impacts on fungal development. In the study by Long et al., cultivable bacteria associated with the spores of *Gigaspora margarita* were isolated. The majority of isolates were affiliated with Proteobacteria, Actinobacteria, and Firmicutes. Of the bacteria identified, 30.2% promoted and 11.6% inhibited spore germination. Six chitin-decomposing bacteria were also isolated from *G. margarita*, a potential mechanism for inducing spore germination [55]. *Paenibacillus validus* isolates in co-culture with *Glomus intraradices* stimulate hyphal growth and promote formation and maturation of spores [56].

Bacteria associated to AMF (*Funneliformis caledonium*, *Racocetra alborosea* and *Funneliformis mosseae*) spore wall were isolated from coastal reclamation land [28]. Bacteria belonging to phyla spanning Firmicutes, Proteobacteria, Actinobacteria, and Bacteroidetes were isolated. To characterize association characteristics, bacteria were screened for production of chitinases, proteases, cellulase enzymes and exopolysaccharide. At least one of these characteristics were found in 113 out of 120 associated bacteria.

In many agriculturally relevant BFIs, the fungal partners differentially produce mycotoxigenic compounds when interacting with their respective bacterial partners. This is particularly true for the *Fusarium* species of fungi, as has been reviewed earlier [57]. These mycotoxins have important ecological roles in securing the niche for the producing microbes by contributing to competitive fitness and promoting virulence characteristics for successful infection and subsequent dispersal. Recent research has found that these observations also hold true for the clinical pathogens, *A. fumigatus* and *P. aeruginosa* [58]. It is important to know that mycotoxins cause significant harm to human health upon ingestion. Research efforts to control mycotoxin production to ensure food safety can benefit from understanding BFIs in greater detail.

### 3.2. Factors Obtained Via BFIs

The obligate endofungal symbiosis between CaGg and *G. maragarita* enhances the fungal bioenergetic potential and ATP production and detoxification of reactive oxygen species [59]. Genomic data brought to light the extreme nutritional dependence of the bacterium for carbon, nitrogen and phosphorus on the fungus. The bacterium synthesizes vitamin B12, antibiotics, and toxin-resistant molecules which may collectively enhance the fungal ecological fitness [32]. Sweeping changes in the fungal metabolome have been reported in response to presence of the endobacteria, corroborating the genetic findings. The biological relevance of such extensive rewiring of metabolome remains to be studied [60].

Bacteria isolated from the surface-decontaminated spores of AM fungi, *Glomus intraradices* and *Glomus mosseae*, from field rhizospheres samples of *Festuca ovina* and *Leucanthemum vulgare* show antifungal characteristics [61]. Transcriptomic analyses studying the interaction between endobacteria and *S. indica* show the bacteria may produce antibiotics and may be involved in resistance to various antimicrobial compounds [42]. These reports suggest that endobacteria may promote competitive fitness of the fungal host in polymicrobial environments.

### 3.3. Fungal Highways for Bacterial Dissemination

Numerous emerging reports show bacterial movement along hyphae in various niches [62,63], influenced by the physicochemical properties of the microhabitats, in addition to the inherent characteristics of the BFIs [64]. In most cases, bacteria show chemotaxis towards the fungus and employ flagellar-mediated motility for movement, with minor roles for T3 and T4SS. Migration along hyphae can confer competitive fitness advantages [64,65]. The contaminant biodegrading bacterium, *Pseudomonas putida*, showed chemotaxis towards *Pythium ultimum* (an oomycete but with similar hyphal morphology of fungi), and only moved across airspaces in unsaturated soil when present with the oomycete [66], suggesting that such interactions offer novel solutions to deal with anthropogenic contaminants [67,68,69]. *Serratia* sp. move, spread over and kill hyphae of certain fungi. This dispersal mechanism was specific to zygomycetes only, which has been attributed to fungal topography and architecture [17]. *Burkholderia* spp., *Ralstonia* spp., *Dyella* spp., and *Sphingobacterium* spp. all showed migration on the hyphae of the fungus *Lyophyllum* sp. Strain Karsten [70]. *Burkholderia terrae*, that migrates on diverse Asco- and Basidiomycete fungal hosts [71], facilitated the hitch-hiking by *Dyella japonicum* BS013 on the fungal hyphae [72]. Dispersal of *Paraburkholderia terrae* on the Lyophyllum sp. was significantly higher at low pH, suggesting that associations with the fungus may have protective roles [73]. Motile Proteobacteria swim on aqueous films formed on fungal hyphae in cheese rinds, altering cheese rind microbiomes preferentially towards motile bacteria [74]. This confirmed the long-standing hypothesis that water films on hyphal surfaces facilitate bacterial movement, as aerial hyphae cannot support bacterial migration [64]. *B. subtilis* requires its flagella to travel back and forth along the mycelium of *A. nidulans* and promotes hyphal growth through the production of thiamine [62]. With the improvement in microscopic techniques, along with novel in situ methods, many more BFIs in diverse niches are sure to be identified in the near future [75].

## 4. Implications of BFIs on Host Health and Disease

The previous sections of this review have identified mechanisms of bacterial attachment and invasion into fungal structures, as well as demonstrated the impact that these interactions can have on the fitness and dissemination of microbial partners in BFIs. Oftentimes, these BFIs can also impact either their mammalian or plant host through increased virulence resulting in more severe disease outcomes or by promoting environmental changes that positively affect other organisms like plants in the vicinity. Here, we highlight the roles of some BFIs in human and plant health.

### 4.1. Humans

BFIs in human health are a relatively understudied area of research. Much of the focus has been on identifying the microorganisms that are a part of the human normal flora and determining the changes that occur in these populations during disease, a movement that was catalyzed by the NIH Human Microbiome Project [4]. The microbiome of healthy individuals is comprised of bacteria, fungi, and viruses that relay protection by educating the host immune response and microbial competition to limit infection by pathogenic microorganisms. Specific BFIs have been identified in several clinical fields, including oral [5,7,9,10], skin [76], and respiratory health [13]. Other implications of BFIs have been found in clinical conditions like sepsis [77]. Some of the most common BFIs identified in human health are either of *C. albicans* and *S. aureus* or *A. fumigatus* and *P. aeruginosa*, of which there are several reviews that discuss their prevalence and overall impact in clinical settings, in addition to other clinically relevant pairings [78,79,80,81,82]. Additionally, there have been several studies that have been reviewed regarding BFI secreted factors and their implications in growth, virulence, and survival of the microorganisms [82,83,84]. However, studies focusing on the direct impact of the physical interactions in in vivo disease models are limited. The best studied physical interactions in vivo are of *C. albicans* and *S. aureus* as described above.

*S. aureus* is a prime example of the impact of BFIs on the invasion and dissemination of pathogens in mammalian disease. *S. aureus* is a part of the human normal flora of skin, nares, and gastrointestinal tract, though it can cause disease when the host is immunocompromised. The fungal opportunistic pathogen *C. albicans* is frequently isolated alongside *S. aureus* forming complex inter-kingdom biofilms on medical equipment and in clinical disorders like cystic fibrosis and bloodstream infections. There have been several studies that have found increased morbidity and mortality when mice are co-inoculated with both organisms [11,77]. In a recent intra-abdominal infection study, 80–100% of mice infected with both *C. albicans* and *S. aureus* were deceased within 20hr post inoculation while individually infected mice did not die in this timeframe. This deadly outcome was attributed to *C. albicans* inducing increased alpha-toxin by *S. aureus* [77]. A similar study using an oral infection model also saw increased morbidity and mortality associated when these two organisms were co-inoculated. Though the finding in this study was that *C. albicans’* invasive hyphae are utilized via physical interactions by *S. aureus* to invade and disseminate within the host tissue and bloodstream. If *S. aureus* is unable to bind to the hyphae, it is no longer able to cause an invasive infection [11].

While *A. fumigatus* and *P. aeruginosa* have been rarely studied using in vivo systems [58,85,86], in vitro data has demonstrated that *P. aeruginosa* binds to the surface of *A. fumigatus* [14]. Clinical and the few in vivo studies available indicate that co-infection with both organisms leads to increased morbidity and mortality [13,58,85,86], much like what is seen with *C. albicans* and *S. aureus* [11,77]. Another aspect to consider when it comes to studying these organisms in vivo is strain variability. Reece et al. found that death of *Galleria mellonella* co-infected with *A. fumigatus* and *P. aeruginosa* occurs in a strain-dependent manner [58]. As these two organisms are commonly isolated with one another, understanding the relevance of physical interactions between the organisms could prove to be enlightening when it comes to their dissemination in the lung.

Our knowledge of these microbial interactions has become especially important in light of the current global pandemic caused by the severe acute respiratory syndrome coronavirus 2 (SARS-CoV2). There is an increased incidence of *Aspergillus* sp. and numerous opportunistic bacteria found in the respiratory secretions of individuals that are infected with SARS-CoV2 [87,88,89]. The risk of developing secondary infections caused by these microorganisms is likely confounded by patients being placed on ventilator support [88]. As we have discussed in previous sections, interactions between bacteria and fungi can result in changes in susceptibility to antibiotics and altered virulence of the organisms. Because there is so little known about SARS-CoV2, we do not currently know what impact that the polymicrobial environment has on the viral response to therapeutics and disease progression. Therefore, study of the virus with other microorganisms should be considered for future research directions.

### 4.2. Plants

As microbes co-exist in the rhizosphere, they form intimate associations with each other bearing consequences for microbial community structure, which in turn influences plant health. Furthermore, as discussed in Section 3.1, bacterial partners can finely tune sporulation, thus altering dissemination of infection propagules, vertical transmission of endofungal bacteria in fungal spores’ aids in their long-distance dispersal, and reprogramming of secondary metabolism results in production of compounds which exacerbate disease development.

‘Biocontrol’ of plant pathogens by utilizing antagonistic microbes that act via antibiosis or direct competition have been very well studied. Perhaps the most famous are the application of *P. fluorescens* and *Trichoderma viridae*. Although biocontrol effects can be clearly identified in vitro, it often fails to provide results in natural settings. These have been thoroughly reviewed [90,91] and will not be discussed here.

In many cases, co-infection results in increased disease severity. One of the earliest reports is of natural field isolates of bacteria associated with *Stagonospora nodorum*, contributing to the increased pathogenicity of the fungus, the causal agent of wheat blotch disease [92]. As discussed earlier, in the famous interaction between *M. rhizoxinica* and *R. microsporus*, rhizoxin production by the bacterium is required for fungal pathogenicity [3]. Bacterial isolates that were co-isolated with *Rhizoctonia solani*, a notorious root rotter were reported to adopt endohyphal behavior in vitro, with a change in morphology associated with the intracellular lifestyle. Colonization of the fungus by these bacteria, belonging to the genus *Enterobacter*, was required for full virulence of the fungus on creeping bentgrass [93]. Co-inoculation of tomato plants with *S. indica* and *Trinickia* sp. T12-10 showed a significant reduction in disease when challenged with the plant wilt pathogen *Fusarium oxysporum* and showed a significantly higher mitigation from disease when infected with *Rhizoctonia solani* [24]. *Burkholderia glumae* and *Fusarium graminearum* frequently co-isolated from infected rice grains showed physical attachment in in vitro assays. Co-infection of rice seedlings showed an increase in disease severity and deoxynivalenol production, which is a virulence factor for *F. graminearum* [44]. Deoxynivalenol, also called vomit toxin, can cause acute vomiting, abdominal pain, and fever [94]. Thus, this BFI not only increases yield loss threatening food security, but also threatens food safety.

On the contrary, certain BFIs also promote nutrient acquisition by facilitating plant associations with mycorrhizal fungi and nitrogen fixing bacteria to boost plant health. In addition to protection from disease, BFIs between plant-helper microbes enhance plant growth. *S. indica* is a plant endophyte known for its plant protecting roles. Addition of *Trinickia* sp. T12-10 to *S. indica* inoculated tomato plants resulted in enhanced fungal colonization of plant roots [24]. Successful endofungal symbiosis between *R. radiobacter* and *S. indica* has been shown to be vital for effective plant growth promotion and systemic resistance against powdery mildew infection in barley [19]. A microscopic analysis by Paul et al. suggests that the basidiomycete yeast endophyte *Rhodotorula mucilaginosa*, in its filamentous form, hosts *Pseudomonas stutzeri* as an endofungal diazotroph. This association is reported to enable the fungus to fix nitrogen and grow in N-deplete conditions. The bacterial dinitrogen reductase required to convert atmospheric dinitrogen gas into ammonia was transcribed by the fungus under adequate N. Co-inoculation of rice plants with both partners was shown to promote plant growth and plant nitrogen nutrition [95]. Interaction between CaGg and *G. margarita* promotes fungal response to strigalactones, which are plant small molecules that stimulate branching and establishment of AMF symbioses [27]. MHB are mycorrhizal helper bacteria defined as bacteria that positively interact with established symbiosis between mycorrhizal fungi and plants, or mycorrhization helper bacteria that facilitate symbiosis between mycorrhizal fungi and plants [46]. They primarily function by stimulating fungal growth and enhancing contact between the fungi and plants. Numerous reports of MHB isolation, identification, and studies on their consequences exist [49,50,51,52,53]. The roles of MHB in symbiosis, their effects on arbuscular mycorrhizal or ectomycorrhizal host, and their specificities and mechanisms have been studied for decades and have been reviewed extensively elsewhere [46,47,48]. Such tri-partite interactions help promote plant growth as well as stress tolerance [47,54,55,56].

## 5. Conclusions

As the field of bacterial-fungal interactions grows, we begin to understand the implications that these interactions have not only on the microorganisms, but also on plant and human hosts. This review shows that there are several areas of overlapping commonalities between BFIs in agriculture and clinical settings, including the use of fungal structures for bacterial dispersal [11,62,63,73], using ECM for bacterial attachment [14,18], and developmental changes that can occur in both fungi and bacteria [43,49] (Figure 2). It is possible that there are even more commonalities that have yet to be discovered, including endosymbiosis in clinical settings. While we focus mainly on the microorganisms involved in these interactions for this review, we would be remiss if we did not mention the potential impact that the host and the niche-specific environmental factors, can have on these interactions as well. Environmental factors include but are not limited to temperature, pH, nutrients, and oxygen levels in the specific niches. These undoubtedly contribute to physical interactions between bacteria and fungi. Comparing and contrasting aspects of microbial interactions in the context of the environment during plant, animal and human hosts interactions would provide a more complete understanding of the mechanisms that facilitate BFI establishment.

While there have been great strides to understanding these interactions in the agricultural setting with model organisms like AM-fungi, clinical settings are still in the initial stages of this research [96]. As many microorganisms in the agricultural setting also have clinical significance, we can begin to reconcile agricultural findings with clinical research. This is especially important now as we begin to explore novel SARS-CoV2 and its interaction with polymicrobial communities in the lung that are made up of bacterial and fungal organisms [97].

## Figures and Tables

**Figure 1 jof-06-00243-f001:**
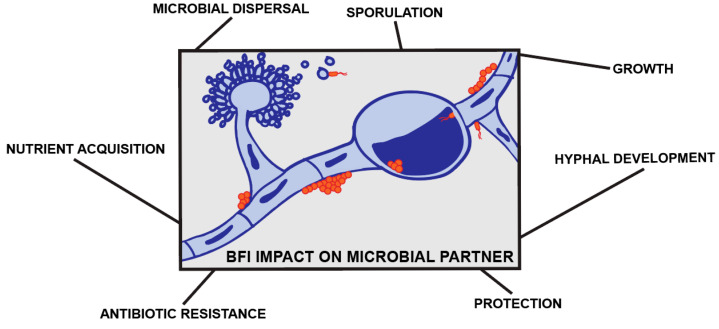
Bacterial-fungal interactions (BFI) impact on microbial partners. Bacterial (orange) and fungal (blue) interactions have impacts on each other’s growth and development in addition to competitive advantages associated with protection, dispersal, and nutrient acquisition.

**Figure 2 jof-06-00243-f002:**
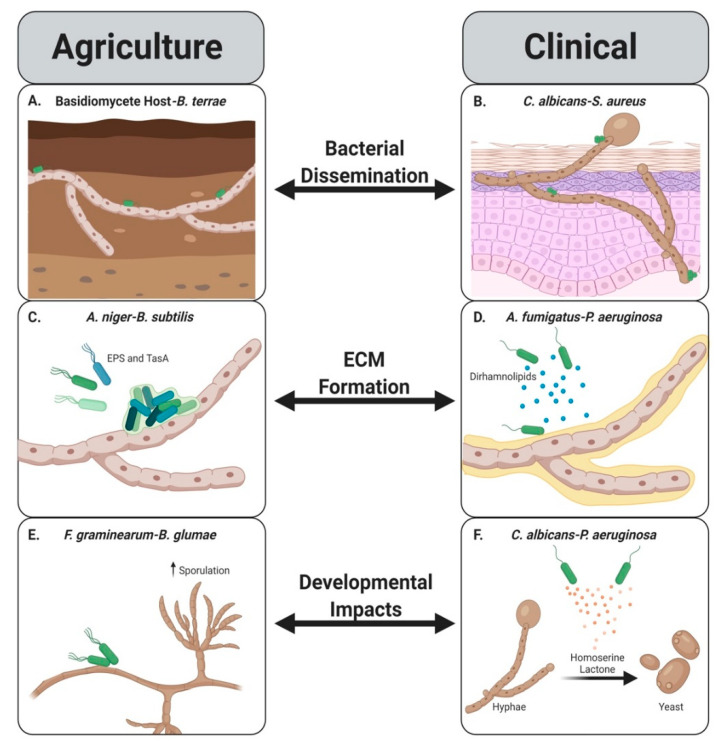
Common outcomes of BFIs in agriculture and clinical systems. Bacteria (green) travel along fungal hyphae (tan) (**A**,**B**) resulting in their dissemination. The ECM produced by bacteria ((**C**)—green, EPS and TasA) or induced ECM (yellow) by bacterial mediators ((**D**)—Blue, dirhamnolipids) mediates bacterial binding to the fungal organisms (tan) (**C**,**D**). Developmental changes occur in fungi (tan) upon interaction with bacteria (green) (**E**,**F**). Created with BioRender.com.

**Table 1 jof-06-00243-t001:** Mediators of Bacterial-Fungal Interactions (BFIs).

Mediators	BFI	Citation
***Surface molecules***		
Als Proteins and *O*-mannosylations	*S. epidermidis-C. albicans*	Beaussart, 2013
Als3p	*S. gordonii-C. albicans* *S. aureus-C. albicans*	Silverman, 2010 Peters, 2012
Als adhesins, SasF, Atl	*S. aureus-C. albicans*	Schlecht, 2015
CbpD	*P. aeruginosa-C. albicans*	Ovchinnikova, 2013
T2SS	*M. rhizoxinica-R. microsporus*	Moebius, 2014
T3SS	*M. rhizoxinica-R. microsporus*	Lackner, 2011
T2SS, T3SS, T4SS, gspD, sec system	*M. rhizoxinica-R. microsporus*	Ghignone, 2012
TAL effector	*M. rhizoxinica-R. microsporus*	Richter, 2020; Carter, 2020
***Genes***		
*gspD*, *secB*	*Ca.* G. gigasporum-*G. margarita*	Ghignone, 2011
*vacB*	*Burkholderia* sp.-*G. margarita*	Ruiz-Lozano & Bonfante, 2000
*spo0A*	*B. subtilis-A. niger* *B. subtilis-Ag. bisporus*	Kjeldgaard, 2019
***Secreted Factors***		
EPS and TasA	*B. subtilis-A. niger* *B. subtilis-A. bisporus*	Kjeldgaard, 2019
GAG, pyo-melanin, 1,8-dihydroxynaphthalene- melanin	*P. aeruginosa-A. fumigatus*	Briard, 2017
ECM	*S. aureus- C. albicans*	Harriot and Noverr, 2009
Holrhizin A	*M. rhizoxinica-R. microsporus*	Moebius, 2014
Ralsolamycin	*R. solanacearum-A. flavus*	Spraker, 2016
Progidiosin, T6SS, TssJ, murein lipoprotein	*S. marcescens-M. irregularis*	Hazarika, 2020
***Other***		
Acid-Base Attractive Forces	*P. aeruginosa-C. albicans*	Ovchinnikova, 2012
Fungal Viability	*B. cereus-Glomus* sp. *P. peoriae-Glomus* sp. *P. brasilensis-Glomus* sp. *S. marcescens-A. fumigatus* *S. marcescents-R. oryzae*	Toljander, 2006 Hover, 2016

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
