# Peer review of "Let’s Get Physical: Bacterial-Fungal Interactions and Their Consequences in Agriculture and Health"

_jof, 2020, doi:10.3390/jof6040243_

Round 1

Reviewer 1 Report

The present review on bacterial-fungal interactions and their consequences in agriculture and health aims to bridge two usually separated field of research. The authors describe fungal-bacterial interactions typically found in humans and in plants/soil and try to link them together. While they collected a lot of information on fungal-bacterial interactions in plants/soil, they presented less information on similar phenomena in the human body.

I really appreciate the idea behind this review and would therefore like to suggest to following alterations in order to improve the accessibility and impact of the review.

MAJOR COMMENTS

1) For certain aspects of the broad field of fungal-bacterial interactions, referring to other researchers’ reviews seems like a very good idea; especially regarding human pathogenicity. It is impossible to cover all the literature in one review, of course, so naming some really helpful reviews will be useful for the interested reader.

2) I really like the idea of bridging human and plant pathogenicity. But I think in its current state, the review is mostly a literature collection. Most prominently for me, it lacks supplementing information for the reader having a stronger background in only one field or the other (or none). So I suggest that the authors compare the different hosts/niches shortly side-by-side (pH, major carbon and nitrogen source, flow, humidity, …). This is especially relevant since these conditions impact also the fungal-bacterial interaction. Best would be a drawing, but a table could also work. In addition, a short introduction of typical plant hosts as well as plant and human pathogens/symbionts would be nice to have as an overview for the reader new to the field/topic. Also here, a graphical representation would likely be the most accessible way.

3) In most sections of the review, there is hardly any information on the experimental conditions of the mentioned experiments. I suggest that the authors add more information because the experimental conditions have a strong impact on the outcome of fungal-bacterial interactions.

4) Unfortunately, some parts of the review are still quite difficult to read since sentences are long and wordy. Please try to consider this when revising the manuscript.

5) The conclusion could emphasize the similarities and differences of fungal-bacterial interactions in humans and plants/soil even more and the potential common ground and synergies between the fields. Where are the benefits of checking what others are finding.

MINOR COMMENTS

Line 20: capital K in kingdom

Line 31: Please specify what this refers to. Please distinguish between commonly manifest and commonly studied. “BFIs commonly manifest as biofilm”.

Line 33/34: Unclear connection “production of agonistic and antagonistic agents

34 (clinical drugs, fungicides, prebiotics etc.)”

Line 53: Picture quality seems quite poor in my version of the manuscript. Also, naming some bacteria and fungi might be increasing the impact of this relatively simple drawing.

Line 62: C.a. not italic; also seen in line 65 and other sections

Line 62: Major risk factors for candidiasis include also dysbiosis.

Line 63: It is not correct to refer to Candida filaments as mold. Please use the term “hypha/hyphae” instead. And make sure to use the right singular/plural form throughout the article. Questionable for instance in line 66, 69, 80, and others.

Line 64: ssp.

Line 81: Unclear reference

Line 88: Check plural

Line 101: Reference needed

Line 121: Capital E

Line 125: Better use “live” instead of “living”

Line 144: THE missing?

Line 145: Unclear

Line 146: is?

Line 177: has?

Line 182: symbiosis?

Lien 195: In vitro italic, please check entire article

Line 216: 2.2.1 very hard to read, structure could be improved

Line 250: Difficult sentence

Line 257: Misleading black lines, maybe better remove?

Line 313: Consider not starting sentence with number.

Line 381: Strong oversimplification. Please refer to other reviews to guide the reader. There are much more fungal-bacterial interactions described in the literature.

Line 390: CF is more commonly referred to as disorder, not disease.

Line 402: Has demonstrated, not found

Author Response

The present review on bacterial-fungal interactions and their consequences in agriculture and health aims to bridge two usually separated field of research. The authors describe fungal-bacterial interactions typically found in humans and in plants/soil and try to link them together. While they collected a lot of information on fungal-bacterial interactions in plants/soil, they presented less information on similar phenomena in the human body.

I really appreciate the idea behind this review and would therefore like to suggest to following alterations in order to improve the accessibility and impact of the review.

Thank you for your kind words.  We do want to note that there is less literature on fungal-bacterial interactions in the body but we think we have met the concern through revision and new figure 2.

 MAJOR COMMENTS

1) For certain aspects of the broad field of fungal-bacterial interactions, referring to other researchers’ reviews seems like a very good idea; especially regarding human pathogenicity. It is impossible to cover all the literature in one review, of course, so naming some really helpful reviews will be useful for the interested reader.

Thank you for this suggestion, we have included several reviews that could be helpful for the interested readers (including lines 382-387).

2) I really like the idea of bridging human and plant pathogenicity. But I think in its current state, the review is mostly a literature collection. Most prominently for me, it lacks supplementing information for the reader having a stronger background in only one field or the other (or none). So I suggest that the authors compare the different hosts/niches shortly side-by-side (pH, major carbon and nitrogen source, flow, humidity, …). This is especially relevant since these conditions impact also the fungal-bacterial interaction. Best would be a drawing, but a table could also work. In addition, a short introduction of typical plant hosts as well as plant and human pathogens/symbionts would be nice to have as an overview for the reader new to the field/topic. Also here, a graphical representation would likely be the most accessible way.

Thank you for this insightful suggestion. We have pointed out a few commonalities between bacterial-fungal interactions in agricultural and human hosts, in the form of a figure (Figure 2). While we do acknowledge the importance of plant/human hosts in our conclusion and discuss implications to such hosts in the text, the review primarily deals with bacterial-fungal partners. The suggested table would be a creative way to discuss host factors, no doubt, but is out of the scope of this review. Including a table comparing and contrasting the host factors would need us to extend the review to include the response of hosts to these interactions and, frankly, there are of as yet almost no studies of this sort in the medical field. We will be happy to keep this in mind for our future work.

3) In most sections of the review, there is hardly any information on the experimental conditions of the mentioned experiments. I suggest that the authors add more information because the experimental conditions have a strong impact on the outcome of fungal-bacterial interactions.

Thank you for this suggestion. We do include some additional commentary on the importance of looking at different strains in the context of A. fumigatus (lines 408-410). While we agree that experimental conditions can have a major impact on the outcome of these interactions, it is not feasible to include them in this review. We cover several different studies that vary greatly in experimental design. As such it is difficult to compare experimental conditions and it could be misleading until there are more consistent studies on these attachments in the same model organisms.

4) Unfortunately, some parts of the review are still quite difficult to read since sentences are long and wordy. Please try to consider this when revising the manuscript. 

We have considered this and rephrased sentences to offer clarity.

5) The conclusion could emphasize the similarities and differences of fungal-bacterial interactions in humans and plants/soil even more and the potential common ground and synergies between the fields. Where are the benefits of checking what others are finding.

Thank you for this suggestion. In our new figure 2, we have sought to bridge human and agriculture BFIs by showing commonalities in bacterial dispersal on fungal hyphae, ECM production that facilitates bacterial attachment, and changes to the development of microbial partners in BFIs. We have also addressed this in the text of the conclusion section.

 MINOR COMMENTS

 Line 20: capital K in kingdom

Addressed

Line 31: Please specify what this refers to. Please distinguish between commonly manifest and commonly studied. “BFIs commonly manifest as biofilm”.

Addressed

Line 33/34: Unclear connection “production of agonistic and antagonistic agents This has been rephrased.

34 (clinical drugs, fungicides, prebiotics etc.)”

Addressed

Line 53: Picture quality seems quite poor in my version of the manuscript. Also, naming some bacteria and fungi might be increasing the impact of this relatively simple drawing.

We have removed this figure. Later in the review we have included a new figure that shows commonalities between BFIs in ag and clinical settings (now listed as figure 2).

Line 62: C.a. not italic; also seen in line 65 and other sections

Addressed

Line 62: Major risk factors for candidiasis include also dysbiosis.

Addressed

Line 63: It is not correct to refer to Candida filaments as mold. Please use the term “hypha/hyphae” instead. And make sure to use the right singular/plural form throughout the article. Questionable for instance in line 66, 69, 80, and others. Addressed

 Line 64: ssp.

Changed to spp. to follow scientific guidelines

 Line 81: Unclear reference Addressed

 Line 88: Check plural Addressed

 Line 101: Reference needed Addressed

 Line 121: Capital E Addressed

 Line 125: Better use “live” instead of “living” Addressed

 Line 144: THE missing? Addressed

 Line 145: Unclear Addressed

 Line 146: is? Addressed

 Line 177: has? Addressed

 Line 182: symbiosis? Correct as written

 Lien 195: In vitro italic, please check entire article Addressed

 Line 216: 2.2.1 very hard to read, structure could be improved This section has been broken down into three sub-sections, each attributed to a specific example.

 Libe 250: Difficult sentence. The sentence has been reorganized and rephrased

 Line 257: Misleading black lines, maybe better remove? We are unsure what the reviewer meant here so we have not changed anything.

 Line 313: Consider not starting sentence with number. Addressed

 Line 381: Strong oversimplification. Please refer to other reviews to guide the reader. There are much more fungal-bacterial interactions described in the literature. We have included references to other reviews that provide more information regarding fungal-bacterial interactions in human health.

 Line 390: CF is more commonly referred to as disorder, not disease. Addressed

 Line 402: Has demonstrated, not found Addressed

Reviewer 2 Report

The review by Steffan and colleagues is an updated compilation of the current knowledge in a field, which is considered a crucial hot spot of microbiology: bacterial and fungal interactions. There are already some relatively recent reviews on this issue (for example: Deveau et al, FEMS; Pawlowska et al Annual review of Plant Pathology.2018), but here the Authors offer  a very careful revision of BFIs in the context of human health as well of environment, considering all these fascinating interkingdom interactions (humans/plants with fungi and bacteria). The Authors look for the mediators of bacteria-fungal interactions, summarising the main findings in a useful table. In addition, interesting implications on host health and disease are introduced, with novel suggestions on individuals infected by COVID.

I have done some minor remarks.

Table 1 has to be checked, since there are some wrong references. Bianciotto 2004 (as Bianciotto et al in the list?) is probably wrong, as well as the BFI couple which should be R. microsporus. The correct references could be Lackner et al 2011 J. Bacteriol or Lackner, et al 2011, BMC Genomics (cf with references reported at line 178-179). In the context of mediators, an interesting genomic element is given by the active presence of a toxin-antitoxin system in Candidatus glomeribacter (Salvioli et al 2017),

Line 109: "recent " should be deleted. In addition the ref (14) has to be checked

Line 139: Alginate-like EPS of P. fluorescens have been described as involved in vitro in the adhesion between the bacterium and   the surface of AM fungi (Bianciotto et al, 2001).

Line 160: "Mucormycota" to be replaced by Mucoromycota. The authors should probably consider the new taxonomic rules according to Spatafora et al 2016. According to this view (but there are alternative options), Mucoromycota is a phylum which contains different subphyla, among which Glomeromycotina. Looking at the endobacteria, this makes sense; since Mortierella and Rhizopus, which contain obligate endobacteria together with Glomeromycotina, belong to the other subphyla (Mucoromycotina and Mortierellomycotina). The taxonomic relationship between fungal hosts and obligate endobacteria is explained in Bonfante and Venice (2020).

Line 181-182: the sentence should be checked, since for me "solubilize nutrients" is not clear. In addition, as inner structures (they are inside the cortical cells) arbuscules are not involved in the uptake of nutrients from soil.

210: endo-bacterium: please check the writing along the text. Endobacterium (and not endo-bacterium) is the term which is usually used by the community working on endobacteria.

242: "Ectomycorrhizae" has to be replaced by ectomycorrhizal fungi,

         "saprophytic" by saprotrophic

245: I fully agree with this comment; please note that probably this is mostly due to the multiple and diverse interactions which are established between bacteria and fungi and also on some wrong identification. For example the authors of ref 41 speak about endobacterial communities (!), while they are describing the microbiota which is associated to some fungal fruit bodies.

435 please check the sentence

455: what is the reference for the yeast, which contains Pseudomonas? I am wondering whether it is: Paul, Saha, et.al 2020, who investigated the Improvement of nitrogen nutrition in rice by interaction with the basidiomycete Rhodotorula mucilaginosa and its N2-fixing endobacteria. Plant Cell https://doi.org/10.1105/tpc.19.00385. It is a fantastic paper, but unfortunately the localization of Pseudomonas inside the yeast is not so clear! Only a TEM picture may give a convincing demonstration.

290-305. The whole paragraph would require an improving of the rational, since it sounds as a list of references. The BFIs leading to an increase in fungal sporulation could be put together quoting and commenting Mondo et al 2017 Nature Communications.

In order to avoid confusion in the reader, the bacteria - introduced at line 305- should be identified as cultivable (just to distinguish them from the endobacteria, which are usually not-cultivable, due to their reduced genome.

Paragraph 32. Also here, I would suggest linking the sentences. The examples given at line 336 and 337 refer to endobacteria and would go to line 332-333, while again the bacteria introduced at line 334 are bacteria living at the surface and cultivable..They could be moved to the previous paragraph.

Author Response

The review by Steffan and colleagues is an updated compilation of the current knowledge in a field, which is considered a crucial hot spot of microbiology: bacterial and fungal interactions. There are already some relatively recent reviews on this issue (for example: Deveau et al, FEMS; Pawlowska et al Annual review of Plant Pathology.2018), but here the Authors offer  a very careful revision of BFIs in the context of human health as well of environment, considering all these fascinating interkingdom interactions (humans/plants with fungi and bacteria). The Authors look for the mediators of bacteria-fungal interactions, summarising the main findings in a useful table. In addition, interesting implications on host health and disease are introduced, with novel suggestions on individuals infected by COVID.

I have done some minor remarks.

Table 1 has to be checked, since there are some wrong references. Bianciotto 2004 (as Bianciotto et al in the list?) is probably wrong, as well as the BFI couple which should be R. microsporus. The correct references could be Lackner et al 2011 J. Bacteriol or Lackner, et al 2011, BMC Genomics (cf with references reported at line 178-179). In the context of mediators, an interesting genomic element is given by the active presence of a toxin-antitoxin system in Candidatus glomeribacter (Salvioli et al 2017),

Thank you for pointing this out. These have been addressed.

Line 109: "recent " should be deleted. In addition the ref (14) has to be checked

Addressed and the reference is correct.

Line 139: Alginate-like EPS of P. fluorescens have been described as involved in vitro in the adhesion between the bacterium and   the surface of AM fungi (Bianciotto et al, 2001).

Unsure as to what the reviewer is suggesting here so we have not changed anything.

Line 160: "Mucormycota" to be replaced by Mucoromycota. The authors should probably consider the new taxonomic rules according to Spatafora et al 2016. According to this view (but there are alternative options), Mucoromycota is a phylum which contains different subphyla, among which Glomeromycotina. Looking at the endobacteria, this makes sense; since Mortierella and Rhizopus, which contain obligate endobacteria together with Glomeromycotina, belong to the other subphyla (Mucoromycotina and Mortierellomycotina). The taxonomic relationship between fungal hosts and obligate endobacteria is explained in Bonfante and Venice (2020).

Thank you for pointing this out. This view has been incorporated into the text (Lines 161-166).

Line 181-182: the sentence should be checked, since for me "solubilize nutrients" is not clear. In addition, as inner structures (they are inside the cortical cells) arbuscules are not involved in the uptake of nutrients from soil.

The sentence has been rephrased to provide clarity.

210: endo-bacterium: please check the writing along the text. Endobacterium (and not endo-bacterium) is the term which is usually used by the community working on endobacteria.

Addressed

242: "Ectomycorrhizae" has to be replaced by ectomycorrhizal fungi, "saprophytic" by saprotrophic

Addressed

245: I fully agree with this comment; please note that probably this is mostly due to the multiple and diverse interactions which are established between bacteria and fungi and also on some wrong identification. For example the authors of ref 41 speak about endobacterial communities (!), while they are describing the microbiota which is associated to some fungal fruit bodies.

Thank you for the comment!

435 please check the sentence.

Sentence has been rephrased for better clarity.

455: what is the reference for the yeast, which contains Pseudomonas? I am wondering whether it is: Paul, Saha, et.al 2020, who investigated the Improvement of nitrogen nutrition in rice by interaction with the basidiomycete Rhodotorula mucilaginosa and its N2-fixing endobacteria. Plant Cell https://doi.org/10.1105/tpc.19.00385. It is a fantastic paper, but unfortunately the localization of Pseudomonas inside the yeast is not so clear! Only a TEM picture may give a convincing demonstration.

Yes, that is the correct reference. We agree with this comment that the endofungal lifestyle has not been confirmed with multiple techniques. Accordingly, the sentence has been rephrased and the reference has been fixed.

290-305. The whole paragraph would require an improving of the rational, since it sounds as a list of references. The BFIs leading to an increase in fungal sporulation could be put together quoting and commenting Mondo et al 2017 Nature Communications.

The paragraph has been rephrased and reorganized and the reference has been incorporated in to text (lines 311-316).

In order to avoid confusion in the reader, the bacteria - introduced at line 305- should be identified as cultivable (just to distinguish them from the endobacteria, which are usually not-cultivable, due to their reduced genome.

Sentence has been rephrased, thank you for pointing this out.

Paragraph 32. Also here, I would suggest linking the sentences. The examples given at line 336 and 337 refer to endobacteria and would go to line 332-333, while again the bacteria introduced at line 334 are bacteria living at the surface and cultivable..They could be moved to the previous paragraph.

All the information on Candidatus Glomeribacter gigasporum and Gigaspora margarita have been consolidated into one paragraph. Bacteria at line 334 (which is line 345 in the revised manuscript) are isolated from surface-decontaminated spores, that suggests that these may be endofungal and hence included in this section.

Round 2

Reviewer 1 Report

General:

I would like to thank the authors for considering all my comments and implementing most of them in a suitable manner. The new figure is very illustrative.

I still stand by my view that the nature of fungal-bacterial interactions does not only depend on the combination of microbes, but also on the surrounding environment. The authors prefer to not dive into this aspect.

Therefore I suggest that the authors use the conclusion (which is now really short) to point this out and make sure that readers are aware of that limitation of the review.

Further, the bridging aspect between fields and its potential should be pointed out again.

Minor: Line 420 - The newly included reviews on fungal-bacterial interactions in humans could be implemented more smoothly

Author Response

I would like to thank the authors for considering all my comments and implementing most of them in a suitable manner. The new figure is very illustrative. I still stand by my view that the nature of fungal-bacterial interactions does not only depend on the combination of microbes, but also on the surrounding environment. The authors prefer to not dive into this aspect. Therefore I suggest that the authors use the conclusion (which is now really short) to point this out and make sure that readers are aware of that limitation of the review.

Thank you for taking the time to review our document. We appreciate your insight into the new figure and we are glad that you find it illustrative. We have followed your suggestion and added more to the conclusion regarding the environmental impact on BFIs. 

Further, the bridging aspect between fields and its potential should be pointed out again.

We address this in the conclusion.

Minor: Line 420 - The newly included reviews on fungal-bacterial interactions in humans could be implemented more smoothly

We have addressed this.